# Application of Grey Lotka-Volterra Model in Water-Economy-Industry-Technology Innovation System in Beijing-Tianjin-Hebei Region

**DOI:** 10.3390/ijerph19158969

**Published:** 2022-07-23

**Authors:** Xiaorui Guo, Lifeng Wu, Meng Wang

**Affiliations:** 1School of Management Engineering and Business, Hebei University of Engineering, Handan 056038, China; 17532559710@163.com (X.G.); wang17831005110@163.com (M.W.); 2Hebei Key Laboratory of Intelligent Water Conservancy, Hebei University of Engineering, Handan 056038, China

**Keywords:** Grey Lotka-Volterra, fractional GM(1,1), entropy weight method, stability calculation, Beijing-Tianjin-Hebei

## Abstract

With the proposal of China’s high-quality development strategy, how to promote regional stability and coordinated development based on a deep understanding of the main contradictions and changes in China’s society has become the focus of research. High-quality development is a brand-new coordinated development concept, which aims to optimize the economic structure, transform the development model, enhance the development momentum, and take innovation as the primary driving force. How to promote the coordinated development of this region has become a hot issue considered by scholars. The Beijing-Tianjin-Hebei region is the capital economic circle of China, and the purpose of this study is to promote the coordinated and stable development of the region. On this premise, this paper firstly adopts the composite Grey Lotka-Volterra (GLV) model and Fractional GM(1,1) (FGM(1,1)) model to research Water Resources system-Economic System-Industrial System-Technology Innovation System in the Beijing-Tianjin-Hebei region. Secondly, by analyzing the research data, it is found that the relationship between the system is very complex, and the stability calculation results are all below 0. Then, the analysis of the research results shows that there is no obvious coordination among the three regions, and they have not yet reached a state of mutual promotion and stable and coordinated development. Finally, four suggestions are put forward for the coordinated development of the Beijing-Tianjin-Hebei region. This can not only provide direction for the future development of the region but also have reference significance for the development of other regions. Further, accelerate the coordination and unity of all factors of production in China and promote China’s development at a deeper and higher level.

## 1. Introduction

China’s economy has shifted from a stage of rapid growth to a stage of high-quality development. High-quality development needs to focus on and explore the vitality and innovation of development based on a deep understanding of the main contradictions and changes in our society, to achieve the coordination and unity of various institutional elements and promote higher-level development [1]. At present, China has overcapacity in high-pollution and low-tech industries such as chemical, steel, and coal. Low-carbon, clean and high-tech industries are underdeveloped, industrial technology innovation is insufficient and industrial structure is unbalanced [2]. The aggravation of resources and environmental constraints and the lack of innovation power are the key factors restricting the high-quality development of China’s industry [3,4]. High-tech industry is the pillar industry of the national economy and social development, which plays a decisive role in promoting the upgrading of urban industrial structure, improving labor productivity and economic benefits [5].

Beijing-Tianjin-Hebei is located in the northern part of the North China Plain, including Beijing, Tianjin, and Hebei. With an area of about 216,000 square kilometers, it is the political and cultural center of China and known as the Capital Economic Circle. Problems such as water shortage and uneven development of scientific and technological innovation capabilities have gradually become problems restricting the development of the Jing-Jin-Ji region. Although the coordinated development of Beijing-Tianjin-Hebei region has initially formed a new situation of mutual benefit in 2020, there are still many problems [6].

In terms of rural and urban development, Yang et al. [7] used a method of coupling and coordination to spatially analyze the relationship between production, life and ecology in rural areas. Furthermore, Han et al. [8] also used a coupling method to analyze the relationship between Economy-Resources-Environment in this region. Tang et al. [9] studied the spatial evolution of urban expansion in this region, and the results of the research provided suggestions for urban expansion and coordinated development in this region. In the field of energy and environmental research, Hu et al. [10] conducted research on industrial energy and related emissions in the Jing-Jin-Ji region. Wang et al. [11] used grey correlation analysis, FGM(1,1) model (the prediction results are affected by fixed order) and grey multivariate AGMC(1,N) model to conduct inter-regional scientific energy prediction analysis for 13 provinces (cities) in 7 different regions of China, which provides a reference for China’s regional energy reform in the short term. Xu et al. [12] analyzed and studied the interaction between greenhouse gases and air pollution in this region, and put forward reasonable opinions. Wu et al. [13] evaluated the coordinated development status of this region from a comprehensive perspective through the design and research of the coordinated development evaluation system of the Beijing-Tianjin-Hebei area, and made up for the lack of a single perspective of the existing results.

Lotka-Volterra model (LV) is a classic model for studying co-evolution between species in natural ecology. Grey Lotka-Volterra (GLV) model was proposed by ecologist Lotka [14] and mathematician Volterra [15] to describe the mutual competition between biological populations. LV model is used by many scholars in different fields, and the model has been optimized and improved. In terms of model improvement, Wu et al. [16] proposed the GLV model based on the grey modeling method, and estimated the parameters of the model by linear programming on the premise of minimum error. Mao et al. [17] introduced fractional derivatives and time delay factors into the GLV model and established the FDGLV model to analyze the synergistic effect between industrial upgrading and urban expansion in Wuhan, then predict the development of the region in the next five years. Mao et al. [18] estimated the delay value through grey delay relationship analysis, and constructed a new grey delay Lotka-Volterra model to analyze the relationship between Wuhan’s R&D and GDP. Anzhelika et al. [19] based on the LV equation, proposed a discrete model with a controllable phase space volume. The model provides almost linear control of the phase space volume by changing the symmetry of the basic finite difference scheme, thereby providing quantitative characteristics of the simulated behavior.

In the application of the model, Mao et al. [20] analyzed the relationship between China’s third-party payment system and online banks through this model. Yuan et al. [21] applied the GLV model to biological leaching of uranium, which has important theoretical and practical significance for improving the synergy of microorganisms in biological leaching of uranium and improving the efficiency of biological leaching of uranium. Wang et al. [22] combined the grey prediction theory with the Lotka-Volterra competition model to explore and analyze the dynamic competition among smart TVs, flat-panel TVs, android and ios smart phone operating systems. Wu et al. [23] used the Grey Lotka-Volterra model to analyze the relationship between Ningbo’s GDP and FDI from 1999 to 2007, and used the discrete LV model to predict Ningbo’s GDP and FDI. Wang et al. [24] proposed a three-dimensional Lotka-Volterra model. Taking the sales volume of three local auto companies in China as a sample, they conducted an empirical analysis of the symbiosis system from the two aspects of the balanced development of the three varieties and the evolution of the competition among the three varieties. Anindya S. Chakrabarti [25] applied the Lotka-Volterra equation to simulate the evolution of the technological frontier, proving that the way of technological diffusion has produced repercussions in partner economies. Meng et al. [26] established the government-port-third-party organization evolutionary game LV model for the development of green smart ports, and made suggestions for the development of green smart ports in the future. Zhang et al. [27] investigated the competition and cooperation between Shaanxi Province’s TI-RE-EE-IQ by constructing a four-dimensional Grey Lotka-Volterra model, and explored the balance and stability of the four-dimensional model.

According to the findings of the literature in this region, most scholars have begun to study the spatial scope of the Beijing-Tianjin-Hebei region, using different methods such as coupled correlation to study the relationship between systems. Although most scholars use coupling degree to evaluate coordination, the evaluation method of coupling degree is easily affected by extreme data. Lotka-Volterra model is mostly used to describe the relationship between biological populations, but rarely used in regional cooperative development. However, multidimensional Lotka-Volterra model is an in-depth study on the basis of Lotka-Volterra model. In recent years, high-quality development and new development concepts have become the basic orientation and guidelines of China’s economic development, and the study of Beijing-Tianjin-Hebei coordinated development from the perspective of high-quality development has attracted academic attention. It has become a consensus to study the coordinated development of Jing-Jin-Ji region from the perspective of high-quality development, but the index system and regional level of existing research are not detailed enough. Therefore, it is of great theoretical and practical significance to construct a multi-system Lotka-Volterra model to study the coordinated development of Beijing-Tianjin-Hebei region from the perspective of high-quality development.

## 2. Data and Materials

### 2.1. Study Areas

The coordinated development of Beijing-Tianjin-Hebei is one of the three key areas of major national strategic breakthroughs. Coordinated development between regions has received active policy support from the state and the Beijing-Tianjin-Hebei local government. Therefore, the Beijing-Tianjin-Hebei region is selected as the research object in this paper, and the location of the selected area is shown in Figure 1.

Beijing is located in the north of China and its center located at 116°20′ east longitude. The terrain northwest high, southeast low. Surrounded by mountains on three sides in the west, north and northeast, the southeast is a plain sloping slowly toward the Bohai Sea and adjacent to Tianjin in the east. By the end of 2021, Beijing had 21.886 million permanent residents.

Tianjin is surrounded by the Bohai Sea in the east, Yanshan Mountains in the north and the capital Beijing in the west. The geographical coordinates are between 38°34′–40°15′ N and 116°43′–118°4′ E. Tianjin is the main node of the China-Mongolia-Russia Economic Corridor, the strategic fulcrum of the Maritime Silk Road, the intersection of the Belt and Road, and the nearest eastern starting point of the Eurasian Land Bridge. It is also a shipping center, logistics center and modern manufacturing base in northern China. By the end of 2021, the city has jurisdiction over 16 districts with a total area of 11,966.45 square kilometers and a permanent population of 13.73 million.

Hebei is located in the north China plain, east of the Bohai Sea, surrounded by Beijing and Tianjin, the Taihang Mountains in the west, Yanshan Mountains in the north, and Zhangbei Plateau in the north. It is the only province in China with plateaus, mountains, hills, plains, lakes and coasts. It is an important grain and cotton producing area in China. By the end of 2021, Hebei had 74.48 million permanent residents.

### 2.2. Data Collection

In this paper, Beijing-Tianjin-Hebei region is taken as the research area, and the coordinated development of the region is studied by constructing a composite Water Resources System-Economic System-Industrial System-Technology Innovation System. Original data of the four systems in the Beijing-Tianjin-Hebei region comes from the statistical yearbooks of the provinces and the statistical bulletins of the provinces. Specific evaluation indexes are shown in Table 1.

## 3. Model Formulation

In this paper, grey Lotka-Volterra model is used to analyze and calculate the relationship between Water Resources System-Economy System-Industrial System-Technology Innovation System in the Beijing-Tianjin-Hebei region. Firstly, the relationship analysis is derived on the current data in the Beijing-Tianjin-Hebei region. Secondly, using the FGM(1,1) model to forecast the existing data. Then, this model analysis is performed on the predicted data. Finally, combining the existing and predicted relationships among the four systems in this region, suggestions for improvement are put forward.

### 3.1. Data Processing

This article uses the entropy method to process the original data, and the processed data is divided according to the system. The specific steps are as follows.

**Step 1:** Index selection. There are t years, where $X_{i}(t)$ represents the i index in year *t*.**Step 2:** Standardization.
(1)$$X_{i}{}^{\prime }(t)=\frac{\left(X_{i}(t)-\min X_{i}\right)}{\left(\max X_{i}-\min X_{i}\right)}$$**Step 3:** The proportion of indicators is determined.
(2)$$Y_{i}=\frac{{X^{\prime }}_{i}(t)}{\sum {X^{\prime }}_{i}(t)}$$**Step 4:** Calculate entropy.
(3)$$e_{i}=-k\sum Y_{i}\ln(Y_{i}),k=\frac{1}{\ln t}$$**Step 5:** Calculate the redundancy of information entropy.
(4)$$d_{i}=1-e_{i}$$**Step 6:** Calculate the weight of each indicator.
(5)$$W_{i}=\frac{d_{i}}{\sum_{i}d_{i}}$$

### 3.2. Forecasting Model

This paper uses the fractional model (FGM(1,1)) to predict data. Using the fractional model to make predictions can weaken the randomness of the original sequence and make the grey prediction model less disturbed. In addition, FGM(1,1) conforms to the principle of giving priority to new information, which improves the reliability of data prediction results [28,29]. The steps of FGM(1,1) are shown in Figure 2.

### 3.3. Calculation Model

#### 3.3.1. Grey Lotka-Volterra Model

Assume that,
(6)$$X^{\left(0\right)}=\left\{x^{\left(0\right)}(1),x^{\left(0\right)}(2),\ldots ,x^{\left(0\right)}(n)\right\},Y^{\left(0\right)}=\left\{y^{\left(0\right)}(1),y^{\left(0\right)}(2),\ldots ,y^{\left(0\right)}(n)\right\},$$
are positive sequences of original data for the state variables of a system, where,
(7)$$X^{\left(1\right)}(k)=\sum_{i=1}^{k}x^{\left(0\right)}(i),k=1,2,\ldots ,n,$$
(8)$$X^{\left(1\right)}=\left\{x^{\left(1\right)}(1),x^{\left(1\right)}(2),\ldots ,x^{\left(1\right)}(n)\right\},$$
(9)$$Y^{\left(1\right)}(k)=\sum_{i=1}^{k}y^{\left(0\right)}(i),k=1,2,\ldots ,n,$$
(10)$$Y^{\left(1\right)}=\left\{y^{\left(1\right)}(1),y^{\left(1\right)}(2),\ldots ,y^{\left(1\right)}(n)\right\}.$$

View $x^{\left(1\right)}(k)$ and $y^{\left(1\right)}(k)$ as research variables in order to explore the long-term relationship between $X^{\left(0\right)}$ and $Y^{\left(0\right)}$. Actually, the data before $x^{\left(0\right)}(k)$ often exert an influence on $x^{\left(0\right)}(k)$ and $y^{\left(0\right)}(k),$ that is to say the accumulation of $x^{\left(0\right)}(k)$ will exert an influence on $x^{\left(0\right)}(k)$ and $y^{\left(0\right)}(k).$ According the grey modeling method, at time *k*, the grey derivative of the $X^{\left(1\right)}$ is,
(11)$$dx(k)=x^{\left(1\right)}(k+1)-x^{\left(1\right)}(k),k=1,2,\ldots ,n-1.$$

The mean generated sequence of consecutive of $X^{\left(1\right)}$ is $Z_{x}^{\left(1\right)}(k)=\frac{x^{\left(1\right)}(k)+x^{\left(1\right)}(k+1)}{2},$ and the mean generated sequence of consecutive neighbors of $Y^{\left(1\right)}$ is $Z_{y}^{\left(1\right)}(k)=\frac{y^{\left(1\right)}(k)+y^{\left(1\right)}(k+1)}{2},$ where k=1,2,…,n−1. The equation,
(12)$$x^{\left(0\right)}(k+1)\approx a_{1}z_{x}^{\left(1\right)}(k)-b_{1}{\left(z_{x}^{\left(1\right)}(k)\right)}^{2}-c_{1}z_{x}^{\left(1\right)}(k)z_{y}^{\left(1\right)}(k)$$
is called the grey Lotka-Volterra model. Given the error sequence,
(13)$$\varepsilon_{k}=x^{\left(0\right)}(k+1)-a_{1}z_{x}^{\left(1\right)}(k)+b_{1}{\left(z_{x}^{\left(1\right)}(k)\right)}^{2}+c_{1}z_{x}^{\left(1\right)}(k)z_{y}^{\left(1\right)}(k)$$
the least squares estimate of the parameter sequence $a_{1},b_{1}$ and $c_{1}$ of the grey Lotka-Volterra model is given by,
(14)$$\left(\begin{array}{l} {\hat{a}}_{1}\\ {\hat{b}}_{1}\\ {\hat{c}}_{1} \end{array}\right)={\left(B^{T}B\right)}^{-1}B^{T}Y$$
where,
(15)$$Y=\left(\begin{array}{c} x^{\left(0\right)}(2)\\ x^{\left(0\right)}(3)\\ \vdots \\ x^{\left(0\right)}(n) \end{array}\right),B=\left(\begin{array}{ccc} z_{x}^{\left(1\right)}(2) & -{\left[z_{x}^{\left(1\right)}(2)\right]}^{2} & -z_{x}^{\left(1\right)}(2)z_{y}^{\left(1\right)}(2)\\ \begin{array}{c} z_{x}^{\left(1\right)}(3)\\ \vdots \end{array} & \begin{array}{c} -{\left[z_{x}^{\left(1\right)}(3)\right]}^{2}\\ \vdots \end{array} & \begin{array}{c} -z_{x}^{\left(1\right)}(3)z_{y}^{\left(1\right)}(3)\\ \vdots \end{array}\\ z_{x}^{\left(1\right)}(n) & -{\left[z_{x}^{\left(1\right)}(n)\right]}^{2} & -z_{x}^{\left(1\right)}(n)z_{y}^{\left(1\right)}(n) \end{array}\right).$$

To estimate the unknown parameters $a_{1},b_{1}$ and $c_{1}$ under the criterion of the minimization of the MAPE, can set the objective function as:(16)$$\mathrm{Min}:\frac{1}{n-1}\sum_{k=1}^{n-1}\frac{\left|x^{\left(0\right)}(k+1)-a_{1}z_{x}^{\left(1\right)}(k)+b_{1}{\left(z_{x}^{\left(1\right)}(k)\right)}^{2}+c_{1}z_{x}^{\left(1\right)}(k)z_{y}^{\left(1\right)}(k)\right|}{x^{\left(1\right)}(k+1)}$$

Thereby acquiring the estimated values ${\hat{a}}_{1}$, ${\hat{b}}_{1}$ and ${\hat{c}}_{1}$. Then, obtain the estimated values ${\hat{\alpha }}_{1}$, ${\hat{\beta }}_{1}$, and ${\hat{\gamma }}_{1}$ of $\alpha_{1}$, $\beta_{1}$, and $\gamma_{1}$. Lastly, can obtain the discrete Grey Lotka-Volterra model
(17)$${\hat{x}}^{\left(1\right)}(k+1)=\frac{{\hat{\alpha }}_{1}x^{\left(1\right)}(k)}{1+{\hat{\beta }}_{1}x^{\left(1\right)}(k)+{\hat{\gamma }}_{1}y^{\left(1\right)}(k)},k=1,2,\ldots ,n-1$$

The 1-AGO is ${\hat{x}}^{\left(0\right)}(k+1)={\hat{x}}^{\left(1\right)}(k+1)-{\hat{x}}^{\left(1\right)}(k).$

Similarly, can obtain,
(18)$${\hat{y}}^{\left(1\right)}(k+1)=\frac{{\hat{\alpha }}_{2}y^{\left(1\right)}(k)}{1+{\hat{\beta }}_{2}y^{\left(1\right)}(k)+{\hat{\gamma }}_{2}x^{\left(1\right)}(k)},k=1,2,\ldots ,n-1$$

The 1-AGO is ${\hat{y}}^{\left(0\right)}(k+1)={\hat{y}}^{\left(1\right)}(k+1)-{\hat{y}}^{\left(1\right)}(k).$

Can judge the relationship between $X^{\left(1\right)}$ and $Y^{\left(1\right)}$ from the signs of ${\hat{c}}_{1}$ and ${\hat{c}}_{2}$.

#### 3.3.2. Composite Grey Lotka-Volterra Model

The GLV model of 4 populations can be simplified to,
(19)$$\frac{dx_{i}}{dt}=f_{i}(x)=\alpha_{i}x_{i}(t)+\beta_{ii}x_{i}^{2}(t)+\sum_{j=1;j\neq i}^{4}\beta_{ij}x_{i}(t)x_{j}(t),i=1,2,3$$

According to the difference between the parameters βij and $\beta_{ji}$ (i≠j), 7 possible relationships between factor i and j can exist [20,30], as shown in Table 2.

Let:$$\alpha_{i}=lnA_{i},\beta_{ii}=-B_{ii}\frac{ln\alpha_{i}}{\alpha_{i}-1},\beta_{ij}=-B_{ij}\frac{ln\alpha_{i}}{\alpha_{i}-1},(i\neq j;i=1,2,3;j=1,2,3).$$
(20)$$x_{i}(t+1)=\frac{A_{i}x_{i}(t)}{1+B_{ii}x_{i}(t)+\sum_{j=1}^{4}B_{ij}x_{j}(t)}$$
(21)$$\begin{array}{c} x_{i}(t+1)-x_{i}(t)=\alpha_{i}(\frac{x_{i}(t+1)+x_{i}(t)}{2})+\beta_{ii}{\left(\frac{x_{i}(t+1)+x_{i}(t)}{2}\right)}^{2}\\ +\sum_{j=1;j\neq i}^{4}\beta_{ij}(\frac{x_{i}(t+1)+x_{i}(t)}{2})(\frac{x_{j}(t+1)+x_{j}(t)}{2}),i=1,2,3,4 \end{array}$$

Substituting the original data column when *t* = 1, 2, …, *n* − 1, into the above equation, the matrix equation can be obtained:(22)$$X_{i}=D_{i}{\hat{q}}_{i},i=1,2,3,4$$
where,
(23)$$X_{i}={\left[x_{i}(2)-x_{i}(1),x_{i}(3)-x_{i}(2),\cdots ,x_{i}(n)-x_{i}(n-1)\right]}^{T},i=1,2,3,4$$
(24)$${\hat{q}}_{i}={\left[\alpha_{i},\beta_{ii},\beta_{ij}\right]}^{T},i,j=1,2,3,4;i\neq j$$
(25)$$\begin{array}{l} D_{1}=[\begin{array}{cc} \frac{x_{1}(1)+x_{1}(2)}{2} & {\left[\frac{x_{1}(1)+x_{1}(2)}{2}\right]}^{2}\\ \frac{x_{1}(2)+x_{1}(3)}{2} & {\left[\frac{x_{1}(2)+x_{1}(3)}{2}\right]}^{2}\\ \cdots & \cdots \\ \frac{x_{1}(n-1)+x_{1}(n)}{2} & {\left[\frac{x_{1}(n-1)+x_{1}(n)}{2}\right]}^{2} \end{array}\begin{array}{c} \left[\frac{x_{1}(1)+x_{1}(2)}{2}\right]\left[\frac{x_{2}(1)+x_{2}(2)}{2}\right]\\ \left[\frac{x_{1}(2)+x_{1}(3)}{2}\right]\left[\frac{x_{2}(2)+x_{2}(3)}{2}\right]\\ \cdots \\ \left[\frac{x_{1}(n-1)+x_{1}(n)}{2}\right]\left[\frac{x_{2}(n-1)+x_{2}(n)}{2}\right] \end{array}\\ \begin{array}{ll} \left[\frac{x_{1}(1)+x_{1}(2)}{2}\right]\left[\frac{x_{3}(1)+x_{3}(2)}{2}\right] & \left[\frac{x_{1}(1)+x_{1}(2)}{2}\right]\left[\frac{x_{4}(1)+x_{4}(2)}{2}\right]\\ \left[\frac{x_{1}(2)+x_{1}(3)}{2}\right]\left[\frac{x_{3}(2)+x_{3}(3)}{2}\right] & \left[\frac{x_{1}(2)+x_{1}(3)}{2}\right]\left[\frac{x_{4}(2)+x_{4}(3)}{2}\right]\\ \cdots & \cdots \\ \left[\frac{x_{1}(n-1)+x_{1}(n)}{2}\right]\left[\frac{x_{3}(n-1)+x_{3}(n)}{2}\right] & \left[\frac{x_{1}(n-1)+x_{1}(n)}{2}\right]\left[\frac{x_{4}(n-1)+x_{4}(n)}{2}\right] \end{array}] \end{array}$$

Similarly, *D*_2_, *D*_3_ and *D*_4_ can be obtained, and then the least square method can be used to obtain ${\hat{q}}_{i}=\left[{\hat{\alpha }}_{i},{\hat{\beta }}_{ii},{\hat{\beta }}_{ij}\right]={\left(D_{i}^{T}D_{i}\right)}^{-1}D_{i}^{T}X_{i},i=1,2,3,4;j=1,2,3,4;i\neq j$.

By calculation, can obtain the parameter ${\hat{A}}_{i},{\hat{B}}_{ii},{\hat{B}}_{ij}$. Substituting the parameters into equation, the estimated value ${\hat{x}}_{i}(t)$ can be obtained.
(26)$${\hat{x}}_{i}(t+1)=\frac{{\hat{A}}_{i}{\hat{x}}_{i}(t)}{1+{\hat{B}}_{ii}{\hat{x}}_{i}(t)+\sum_{j=1}^{4}{\hat{B}}_{ij}{\hat{x}}_{j}(t)}(i=1,2,3,4;j=1,2,3,4;i\neq j)$$

### 3.4. Equilibrium Analysis

In order to better understand the evolution results of various factors, the equilibrium and stability of the GLV model are analyzed. GLV model can be expressed as,
(27)$$\frac{dx_{i}}{dt}=\alpha_{i}x_{i}(t)+\beta_{ii}x_{i}^{2}(t)+\sum_{j=1,j\neq i}^{4}\beta_{ij}x_{i}(t)x_{j}(t)=0,i=1,2,3,4$$

Therefore, it can be concluded that it has 16 balance points. The balance equation can be written as follows,
(28)$$\begin{array}{l} E_{0}=(0,0,0,0)\\ E_{1}=(-\frac{\alpha_{1}}{\beta_{11}},0,0,0)\\ E_{2}=(0,-\frac{\alpha_{2}}{\beta_{22}},0,0)\\ E_{3}=(0,0,-\frac{\alpha_{3}}{\beta_{33}},0)\\ E_{4}=(0,0,0,-\frac{\alpha_{4}}{\beta_{44}})\\ E_{5}=(\frac{\alpha_{2}\beta_{12}-\alpha_{1}\beta_{22}}{\beta_{11}\beta_{22}-\beta_{12}\beta_{21}},-\frac{\alpha_{2}\beta_{11}-\alpha_{1}\beta_{21}}{\beta_{11}\beta_{22}-\beta_{12}\beta_{21}},0,0)\\ E_{6}=(\frac{\alpha_{3}\beta_{13}-\alpha_{1}\beta_{33}}{\beta_{11}\beta_{33}-\beta_{13}\beta_{31}},0,-\frac{\alpha_{3}\beta_{11}-\alpha_{1}\beta_{31}}{\beta_{11}\beta_{33}-\beta_{13}\beta_{31}},0)\\ E_{7}=(\frac{\alpha_{4}\beta_{14}-\alpha_{1}\beta_{44}}{\beta_{11}\beta_{44}-\beta_{14}\beta_{41}},0,0,-\frac{\alpha_{4}\beta_{11}-\alpha_{1}\beta_{41}}{\beta_{11}\beta_{44}-\beta_{14}\beta_{41}})\\ E_{8}=(0,\frac{\alpha_{3}\beta_{23}-\alpha_{2}\beta_{33}}{\beta_{22}\beta_{33}-\beta_{23}\beta_{32}},-\frac{\alpha_{3}\beta_{22}-\alpha_{2}\beta_{32}}{\beta_{22}\beta_{33}-\beta_{23}\beta_{32}},0)\\ E_{9}=(0,\frac{\alpha_{4}\beta_{24}-\alpha_{2}\beta_{44}}{\beta_{22}\beta_{44}-\beta_{24}\beta_{42}},0,-\frac{\alpha_{4}\beta_{22}-\alpha_{2}\beta_{42}}{\beta_{22}\beta_{44}-\beta_{24}\beta_{42}})\\ E_{10}=(0,0,\frac{\alpha_{4}\beta_{34}-\alpha_{3}\beta_{44}}{\beta_{33}\beta_{44}-\beta_{34}\beta_{43}},-\frac{\alpha_{4}\beta_{33}-\alpha_{3}\beta_{43}}{\beta_{33}\beta_{44}-\beta_{34}\beta_{43}})\\ E_{11}=(-\frac{l_{1}}{l},\frac{l_{2}}{l},-\frac{l_{3}}{l}0)\\ E_{12}=(-\frac{\eta_{1}}{\eta },\frac{\eta_{2}}{\eta }0,-\frac{\eta_{3}}{\eta })\\ E_{13}=(-\frac{\varphi_{1}}{\varphi },0,\frac{\varphi_{2}}{\varphi },-\frac{\varphi_{3}}{\varphi })\\ E_{14}=(0,-\frac{\nu_{1}}{\nu },\frac{\nu_{2}}{\nu },-\frac{\nu_{3}}{\nu })\\ E_{15}=(-\frac{\lambda_{1}}{\lambda },\frac{\lambda_{2}}{\lambda },\frac{\lambda_{3}}{\lambda },-\frac{\lambda_{4}}{\lambda }) \end{array}$$
where,
(29)$$\begin{array}{l} l=|\begin{array}{l} \beta_{11}\\ \beta_{21}\\ \beta_{31} \end{array}\begin{array}{c} \beta_{12}\\ \begin{array}{l} \beta_{22}\\ \beta_{32} \end{array} \end{array}\begin{array}{l} \beta_{13}\\ \beta_{23}\\ \beta_{33} \end{array}|,l_{1}=|\begin{array}{l} \alpha_{1}\\ \alpha_{2}\\ \alpha_{3} \end{array}\begin{array}{c} \beta_{12}\\ \beta_{22}\\ \beta_{32} \end{array}\begin{array}{l} \beta_{13}\\ \beta_{23}\\ \beta_{33} \end{array}|,l_{2}=|\begin{array}{l} \alpha_{1}\\ \alpha_{2}\\ \alpha_{3} \end{array}\begin{array}{c} \beta_{11}\\ \beta_{21}\\ \beta_{31} \end{array}\begin{array}{l} \beta_{13}\\ \beta_{23}\\ \beta_{33} \end{array}|,l_{3}=|\begin{array}{l} \alpha_{1}\\ \alpha_{2}\\ \alpha_{3} \end{array}\begin{array}{c} \beta_{11}\\ \beta_{21}\\ \beta_{31} \end{array}\begin{array}{l} \beta_{12}\\ \beta_{22}\\ \beta_{32} \end{array}|\\ \eta =|\begin{array}{l} \beta_{11}\\ \beta_{21}\\ \beta_{41} \end{array}\begin{array}{c} \beta_{12}\\ \beta_{22}\\ \beta_{42} \end{array}\begin{array}{l} \beta_{14}\\ \beta_{24}\\ \beta_{44} \end{array}|,\eta_{1}=|\begin{array}{l} \alpha_{1}\\ \alpha_{2}\\ \alpha_{4} \end{array}\begin{array}{c} \beta_{12}\\ \beta_{22}\\ \beta_{42} \end{array}\begin{array}{l} \beta_{14}\\ \beta_{24}\\ \beta_{44} \end{array}|,\eta_{2}=|\begin{array}{l} \alpha_{1}\\ \alpha_{2}\\ \alpha_{4} \end{array}\begin{array}{c} \beta_{11}\\ \beta_{21}\\ \beta_{41} \end{array}\begin{array}{l} \beta_{14}\\ \beta_{24}\\ \beta_{44} \end{array}|,\eta_{3}=|\begin{array}{l} \alpha_{1}\\ \alpha_{2}\\ \alpha_{4} \end{array}\begin{array}{c} \beta_{11}\\ \beta_{21}\\ \beta_{41} \end{array}\begin{array}{l} \beta_{12}\\ \beta_{22}\\ \beta_{42} \end{array}|\\ \varphi =|\begin{array}{l} \beta_{11}\\ \beta_{31}\\ \beta_{41} \end{array}\begin{array}{c} \beta_{13}\\ \beta_{33}\\ \beta_{43} \end{array}\begin{array}{l} \beta_{14}\\ \beta_{34}\\ \beta_{44} \end{array}|,\varphi_{1}=|\begin{array}{l} \alpha_{1}\\ \alpha_{3}\\ \alpha_{4} \end{array}\begin{array}{c} \beta_{13}\\ \beta_{33}\\ \beta_{43} \end{array}\begin{array}{l} \beta_{14}\\ \beta_{34}\\ \beta_{44} \end{array}|,\varphi_{2}=|\begin{array}{l} \alpha_{1}\\ \alpha_{3}\\ \alpha_{4} \end{array}\begin{array}{c} \beta_{11}\\ \beta_{31}\\ \beta_{41} \end{array}\begin{array}{l} \beta_{14}\\ \beta_{34}\\ \beta_{44} \end{array}|,\varphi_{3}=|\begin{array}{l} \alpha_{1}\\ \alpha_{3}\\ \alpha_{4} \end{array}\begin{array}{c} \beta_{11}\\ \beta_{31}\\ \beta_{41} \end{array}\begin{array}{l} \beta_{13}\\ \beta_{33}\\ \beta_{43} \end{array}|\\ \nu =|\begin{array}{l} \beta_{22}\\ \beta_{32}\\ \beta_{42} \end{array}\begin{array}{c} \beta_{23}\\ \beta_{33}\\ \beta_{43} \end{array}\begin{array}{l} \beta_{24}\\ \beta_{34}\\ \beta_{44} \end{array}|,\nu_{1}=|\begin{array}{l} \alpha_{2}\\ \alpha_{3}\\ \alpha_{4} \end{array}\begin{array}{c} \beta_{23}\\ \beta_{33}\\ \beta_{43} \end{array}\begin{array}{l} \beta_{24}\\ \beta_{34}\\ \beta_{44} \end{array}|,\nu_{2}=|\begin{array}{l} \alpha_{2}\\ \alpha_{3}\\ \alpha_{4} \end{array}\begin{array}{c} \beta_{22}\\ \beta_{32}\\ \beta_{42} \end{array}\begin{array}{l} \beta_{24}\\ \beta_{34}\\ \beta_{44} \end{array}|,\nu_{3}=|\begin{array}{l} \alpha_{2}\\ \alpha_{3}\\ \alpha_{4} \end{array}\begin{array}{c} \beta_{22}\\ \beta_{32}\\ \beta_{42} \end{array}\begin{array}{l} \beta_{23}\\ \beta_{33}\\ \beta_{43} \end{array}|\\ \lambda =|\begin{array}{l} \beta_{11}\\ \beta_{21}\\ \beta_{31}\\ \beta_{41} \end{array}\begin{array}{c} \beta_{12}\\ \beta_{22}\\ \beta_{32}\\ \beta_{42} \end{array}\begin{array}{c} \beta_{13}\\ \beta_{23}\\ \beta_{33}\\ \beta_{43} \end{array}\begin{array}{c} \beta_{14}\\ \beta_{24}\\ \beta_{34}\\ \beta_{44} \end{array}|,\lambda_{1}=|\begin{array}{c} \beta_{12}\\ \beta_{22}\\ \beta_{32}\\ \beta_{42} \end{array}\begin{array}{c} \beta_{13}\\ \beta_{23}\\ \beta_{33}\\ \beta_{43} \end{array}\begin{array}{c} \beta_{14}\\ \beta_{24}\\ \beta_{34}\\ \beta_{44} \end{array}\begin{array}{c} \alpha_{1}\\ \alpha_{2}\\ \alpha_{3}\\ \alpha_{4} \end{array}|,\lambda_{2}=|\begin{array}{c} \beta_{11}\\ \beta_{21}\\ \beta_{31}\\ \beta_{41} \end{array}\begin{array}{c} \beta_{13}\\ \beta_{23}\\ \beta_{33}\\ \beta_{43} \end{array}\begin{array}{c} \beta_{14}\\ \beta_{24}\\ \beta_{34}\\ \beta_{44} \end{array}\begin{array}{c} \alpha_{1}\\ \alpha_{2}\\ \alpha_{3}\\ \alpha_{4} \end{array}|\\ \lambda_{3}=|\begin{array}{c} \beta_{11}\\ \beta_{21}\\ \beta_{31}\\ \beta_{41} \end{array}\begin{array}{c} \beta_{12}\\ \beta_{22}\\ \beta_{32}\\ \beta_{42} \end{array}\begin{array}{c} \beta_{14}\\ \beta_{24}\\ \beta_{34}\\ \beta_{44} \end{array}\begin{array}{c} \alpha_{1}\\ \alpha_{2}\\ \alpha_{3}\\ \alpha_{4} \end{array}|,\lambda_{4}=|\begin{array}{c} \beta_{11}\\ \beta_{21}\\ \beta_{31}\\ \beta_{41} \end{array}\begin{array}{c} \beta_{12}\\ \beta_{22}\\ \beta_{32}\\ \beta_{42} \end{array}\begin{array}{c} \beta_{13}\\ \beta_{23}\\ \beta_{33}\\ \beta_{43} \end{array}\begin{array}{c} \alpha_{1}\\ \alpha_{2}\\ \alpha_{3}\\ \alpha_{4} \end{array}| \end{array}$$

Not all equilibrium points are stable. It can be judged by calculating the eigenvalues of the Jacobian matrix [31]. When the eigenvalues of the Jacobian matrix are all negative real parts, the equilibrium point E reaches a stable state. The Jacobian matrix is as follows,
(30)$$\Delta =\left[\begin{array}{cccc} \frac{\partial f_{1}}{\partial x_{1}} & \frac{\partial f_{1}}{\partial x_{2}} & \frac{\partial f_{1}}{\partial x_{3}} & \frac{\partial f_{1}}{\partial x_{4}}\\ \frac{\partial f_{2}}{\partial x_{1}} & \frac{\partial f_{2}}{\partial x_{2}} & \frac{\partial f_{2}}{\partial x_{3}} & \frac{\partial f_{2}}{\partial x_{4}}\\ \frac{\partial f_{3}}{\partial x_{1}} & \frac{\partial f_{3}}{\partial x_{2}} & \frac{\partial f_{3}}{\partial x_{3}} & \frac{\partial f_{3}}{\partial x_{4}}\\ \frac{\partial f_{4}}{\partial x_{1}} & \frac{\partial f_{4}}{\partial x_{2}} & \frac{\partial f_{4}}{\partial x_{3}} & \frac{\partial f_{4}}{\partial x_{4}} \end{array}\right](x_{1},x_{2},x_{3},x_{4})\;\begin{array}{c} =[\begin{array}{ll} \alpha_{1}+2\beta_{11}x_{1}+\beta_{12}x_{2}+\beta_{13}x_{3}+\beta_{14}x_{4} & \beta_{12}x_{1}\\ \beta_{21}x_{2} & \alpha_{2}+2\beta_{22}x_{2}+\beta_{21}x_{1}+\beta_{23}x_{3}+\beta_{24}x_{4}\\ \beta_{31}x_{3} & \beta_{32}x_{3}\\ \beta_{41}x_{4} & \beta_{42}x_{4} \end{array}\\ \begin{array}{ll} \beta_{13}x_{1} & \beta_{14}x_{1}\\ \beta_{23}x_{2} & \beta_{24}x_{2}\\ \alpha_{3}+2\beta_{33}x_{3}+\beta_{31}x_{1}+\beta_{32}x_{3}+\beta_{34}x_{4} & \beta_{34}x_{3}\\ \beta_{43}x_{4} & \alpha_{4}+2\beta_{44}x_{4}+\beta_{41}x_{1}+\beta_{42}x_{2}+\beta_{43}x_{3} \end{array}] \end{array}$$

According to the Routh-Hurwitz system stability criterion, the necessary and sufficient condition for the characteristic roots of the system to have a negative real part is that the highest order coefficient of the characteristic equation is greater than 0, and the formulas composed of the coefficients of the characteristic equation are all positive [32]. Assuming that *p*, *q*, *r* and *s* are the coefficients of the third-order, second-order, first-order, and constant terms of the characteristic equation, then,
(31)$$\left\{ \begin{array}{ll} p & =-(\frac{\partial f_{1}}{\partial x_{1}}+\frac{\partial f_{2}}{\partial x_{2}}+\frac{\partial f_{3}}{\partial x_{3}}+\frac{\partial f_{4}}{\partial x_{4}})\\ q & =|\Delta (1:2,1:2)|+|\Delta (\left[13\right],\left[13\right])|+|\Delta (\left[14\right],\left[14\right])|\\ & +|\Delta (\left[23\right],\left[23\right])|+|\Delta (\left[24\right],\left[24\right])|+|\Delta (\left[34\right],\left[34\right])|\\ r & =-(|\Delta (1:3,1:3)|+|\Delta (\left[124\right],\left[124\right])|\\ & +|\Delta (\left[134\right],\left[134\right])|+|\Delta (2:4,2:4)|)\\ s & =|\Delta | \end{array}\right.$$
where |Δ(1:3,1:3)| means calculating the determinant  |1,2,3|.|Δ([124],[124])| means calculating the determinant  |1,2,4|. Others are the same.

According to the above analysis, when p>0,q>0, qr−ps>0,$qrs-ps^{2}>0,$ the equilibrium points E reaches a stable state. otherwise, it is unstable.

## 4. Results and Discussion

### 4.1. Data Results

#### 4.1.1. Prediction and Analysis of Existing Data

Now take the Beijing area as an example, and use the FGM(1,1) model to forecast the original data. It can be seen from Figure 3 that as time grows, Water Resources System (*WR*), Economy System (*ES*), Industrial System (*IS*) and Technology Innovation System (*TI*) are continuously changing with the growth of time, showing an upward trend roughly. Assuming that there is a connection between these four systems, and now calculate and analyze the connection between the four systems.

In Figure 3, each figure is divided into two parts, with known data from 2015 to 2019 and predicted data from 2020 to 2024. Figure 3a shows the change trend of WR in Beijing from 2015 to 2024. Figure 3b shows the change trend of ES in Beijing from 2015 to 2024. Figure 3c shows the change trend of IS in Beijing from 2015 to 2024. Figure 3d shows the change trend of TI in Beijing from 2015 to 2024.

#### 4.1.2. Status Quo of Beijing-Tianjin-Hebei

(1) Beijing area

Based on the original data of the four systems in the Beijing area, the relationship between the four systems in the Beijing area from 2015 to 2019 is now calculated using composite Grey Lotka-Volterra. The expression formula of the calculation results is as follows.
(32)$$\begin{array}{ll} \frac{dWR(t)}{dt} & =0.79003906WR-0.0463867WR^{{}^{2}}+0.00001788WR\\ & \times ES+0.00000191WR\times IS-0.00009537WR\times TI\\ \frac{dES(t)}{dt} & =1.19726563ES+0.00003874ES^{{}^{2}}-0.01611328ES\\ & \times WR+0.00000042ES\times IS-0.00040436ES\times TI\\ \frac{dIS(t)}{dt} & =0.06250000IS-0.00000620IS^{{}^{2}}+0.04296875IS\times WR\\ & +0.00003815IS\times ES-0.00033569IS\times TI\\ \frac{dTI(t)}{dt} & =1.16406250TI-0.00011063TI^{{}^{2}}-0.04394531TI\times WR\\ & +0.00001955TI\times ES+0.00000155TI\times IS \end{array}$$

The competition and cooperation among the four systems in Beijing is detailed in Figure 4.

(2) Tianjin area

According to the data of the four systems in Tianjin from 2015 to 2019, Equation (33) and Figure 5 are obtained through relevant calculations.
(33)$$\begin{array}{ll} \frac{dWR(t)}{dt} & =1.80468750WR+0.04351807WR^{2}-0.00004959WR\\ & \times ES-0.00005177WR\times IS+0.00082397WR\times TI\\ \frac{dES(t)}{dt} & =1.11132813ES+0.00000000ES^{2}+0.02166748ES\\ & \times WR-0.00004460ES\times IS+0.00013733ES\times TI\\ \frac{dIS(t)}{dt} & =1.25585938IS-0.00005405IS^{2}+0.03549194IS\times WR\\ & -0.00001907IS\times ES+0.00039673IS\times TI\\ \frac{dTI(t)}{dt} & =0.54327393TI+0.00038528TI^{2}+0.02306366TI\times WR\\ & -0.00001386TI\times ES-0.00004627TI\times IS \end{array}$$

(3) Hebei area

By combining the above formulas to calculate the 12 indicators of the four systems in Hebei, Equation (34), Figure 6 are obtained.
(34)$$\begin{array}{ll} \frac{dWR(t)}{dt} & =1.52050781WR-0.00848389WR^{2}-0.00002646WR\\ & \times ES+0.00000030WR\times IS+0.00023961WR\times TI\\ \frac{dES(t)}{dt} & =1.45507813ES-0.00001383ES^{2}-0.00454712ES\\ & \times WR+0.00000089ES\times IS+0.00003588ES\times TI\\ \frac{dIS(t)}{dt} & =-0.71093750IS+0.00000108IS^{2}+0.01733398IS\times WR\\ & -0.00011921IS\times ES+0.00003242IS\times TI\\ \frac{dTI(t)}{dt} & =0.75683594TI+0.00024438TI^{2}+0.00531006TI\times WE\\ & -0.00009012TI\times ES+0.00000040TI\times IS \end{array}$$

#### 4.1.3. Equilibrium and Stability

To analyze the equilibrium and stability among Water Resources System, Economy System, Industrial System and Technology Innovation System in these three areas, the parameters ($\alpha_{i},\beta_{ii},\beta_{ij}$) should be estimated first. Second, the 16 equilibrium points (E0–E15) and the equilibrium point stability of the four factors of Water Resources System, Economy System, Industrial System and Technology Innovation System can be calculated.

As mentioned above, the equilibrium point has practical significance only when all four factors exist and are greater than zero [30]. In the equilibrium points E0–E14, no less than 1 factor has a value equal to 0, and only the 4 factors of E15 are all non-zero.

#### 4.1.4. Forecast Data Calculation

Due to the high accuracy of FGM(1,1) prediction, FGM(1,1) is used to forecast data of Beijing-Tianjin-Hebei region in next five years.

After obtaining the forecast data, perform the grey Lotka-Volterra calculation of the composite system on this data, and evaluate the changes in the degree of coordination between the Beijing-Tianjin-Hebei regions in the next five years. The coordinated change trend of the four systems in the future can be obtained by calculation, and a prediction can be made in advance.

(1) Beijing area
(35)$$\begin{array}{ll} \frac{dWR(t)}{dt} & =3.66406250WR-0.04296875WR^{2}+0.00000000WR\\ & \times ES-0.00000346WR\times IS+0.00039673WR\times TI\\ \frac{dES(t)}{dt} & =1.64099121ES+0.00000358ES^{2}-0.04589844ES\\ & \times WR-0.00000325ES\times IS+0.00026321ES\times TI\\ \frac{dIS(t)}{dt} & =1.39160156IS-0.00000343IS^{2}-0.03466797IS\times WR\\ & -0.00000954IS\times ES+0.00038910IS\times TI\\ \frac{dTI(t)}{dt} & =1.22119141TI+0.00010872TI^{2}-0.06591797TI\times WR\\ & +0.00002289TI\times ES-0.00000274TI\times IS \end{array}$$

(2) Tianjin area
(36)$$\begin{array}{ll} \frac{dWR(t)}{dt} & =1.95410156WR-0.97167969WR^{2}-0.00100517WR\\ & \times ES+0.00226927WR\times IS+0.00622559WR\times TI\\ \frac{dES(t)}{dt} & =1.02246094ES-0.00021553ES^{2}-0.71972656ES\\ & \times WR+0.00195122ES\times IS-0.00492859ES\times TI\\ \frac{dIS(t)}{dt} & =1.50390625IS+0.00216007IS^{2}-0.88671875IS\times WR\\ & -0.00072479IS\times ES+0.00230408IS\times TI\\ \frac{dTI(t)}{dt} & =2.34912109TI+0.00790405TI^{2}-1.05126953TI\times WR\\ & -0.00115395TI\times ES+0.00241613TI\times IS \end{array}$$

(3) Hebei area
(37)$$\begin{array}{ll} \frac{dWR(t)}{dt} & =6.14477539WR-0.00488281WR^{2}+0.01002502WR\\ & \times ES+0.00001642WR\times IS-0.04504395WR\times TI\\ \frac{dES(t)}{dt} & =1.72616577ES+0.00070381ES^{2}-0.51635742ES\\ & \times WR+0.00000510ES\times IS+0.00648499ES\times TI\\ \frac{dIS(t)}{dt} & =1.57000732IS+0.00000642IS^{2}-0.44720459IS\times WR\\ & +0.00182343IS\times ES+0.00013733IS\times TI\\ \frac{dTI(t)}{dt} & =1.69279099TI+0.00714779TI^{2}-0.52215195TI\times WR\\ & +0.00057882TI\times ES+0.00000494TI\times IS \end{array}$$

### 4.2. Data Analysis

#### 4.2.1. Analysis on the Status Quo of Beijing-Tianjin-Hebei

(1) Beijing area

Figure 4 shows that there is a predator-predator relationship among Economic System, Industrial System and Technology Innovation System. This phenomenon shows that the current high-tech industry in Beijing still needs to be improved, and it has not attained a state of benign competition with other industries.

In Table 3 that reflects the existence of a predator-predation relationship between the Water Resources System and Economy System. The Water Resources System hinders the development of the Economy System, but the Economy System provides a boost to the Water Resources System. At the same time, the influence coefficient of the Economy System on the Water Resources System is 0.01611328, and the influence coefficient of the Water Resources System on the Economy System is 0.00001788. This shows that the Economy System has a greater impact on the Water Resources System, and the development of the Economic System determines the development of the Water Resources System. In order to achieve healthy competition between the two systems, it is necessary to strictly control the reasonable use of water resources in the Water Resources System under the conditions of rapid economic development.

This shows that there is a mutualistic relationship between Water Resources System and Industrial System. However, it shows that Industrial System controls Water Resources System, and the increase in industrial output value will face an increase in Water Resources System. Under the requirements of high-quality and sustainable development, the total amount of water supply should be controlled within a reasonable range while increasing industrial added value.

From Table 3, the degree of influence of Technology Innovation System on Water Resource System is 0.0439453125, and Water Resource System on Technology Innovation System is 0.0000953674, indicating that there is a competitive relationship between the two. According to the above data, Beijing should increase the investment in technological innovation, and the continuous progress of technological innovation will reduce the total water supply.

It shows that there is a mutualistic between the Economy System and the Industrial System. Based on data, it can be seen that as the total industrial output continues to increase, the economic system also increases.

There is a predator-predator relationship between the Economy System and Technology Innovation System. The impact coefficient of Economy System on Technology Innovation System is greater than the impact of Technology Innovation System on the Economy System, indicating that economic conditions determine the development of technological innovation. Economic foundation determines the ability to innovate. Therefore, to improve the level of local economy, but also pay attention to the development of innovation ability.

Predatory relationship exists between Industrial system and Technology Innovation system. The continuous innovation of science and technology will promote an increase in industrial output value. Thus, paying attention to technological progress, actively adjusting the industrial structure, and eliminating outdated production capacity. By improving industrial efficiency and developing high-tech industries, Beijing’s industrial development will be more in line with the needs of the people.

(2) Tianjin area

It can be seen from Figure 5 that there is a predator-predator relationship among the Tianjin Water Resources System, Economy System and Industrial System. There is a mutual relationship between the Water Resources System and the Technology Innovation System. There is a competitive relationship between the Economy System and Industrial System systems. Economy System, Industrial System and Technology Innovation System have a predatory relationship.

Now analyze it based on the detailed data in Table 4. It indicates the existence of a predation relationship between the system and the system. The Economy System hinders the development of the system, but the Water Resources System provides a boost to the Economy System. The influence coefficient of the Economy System on the Water Resources System is 0.02166748, and the influence coefficient of the Water Resources System on the Economy System is 0.00004959. This shows that the Economy System has a greater impact on the Water Resources System, and the development of the economic system determines the development of water resources.

Predation relationships exist between Water System and Industrial System. The predator-prey relationship means that the development of one system hinders the development of another system. Due to the Industrial System has a greater impact on the Water Resources System. The emergence of this phenomenon shows that the connection between the industrial added value of Tianjin and the water resources system is poor, and the water consumption has not been reasonably controlled under the premise of securing the steady growth of industrial output.

There is a mutual relationship between Water Resources System and Technology Innovation System. The degree of Technology Innovation System’s influence on Water Resources System is 0.02306366, and the degree of Water Resources System’s influence on Technology Innovation System is 0.00082397, reflecting that technological innovation has a greater influence on the total water supply. Although there is a mutually beneficial relationship between the two, they have not reached a competitive relationship. Only proper competitive relations can foster the high-quality development of various indicators.

Economy System and Industrial System are competing shows in Figure 5. The current economic situation of Tianjin is closely associated with gross industrial output value. To improve the regional economy, it is necessary to grasp the value of industrial output value, but the influence degree between them and the relationship between them still need to be stable.

Predator-predator relationship between Economy System and Technology Innovation System. The effect coefficient of Economy System on Technology Innovation System is greater than the impact of Technology Innovation System on Economy System, indicating that economic conditions determine the development of technological innovation. Economic base determines innovation ability. Therefore, to improve the level of local economy, but also pay attention to the development of innovation ability.

Obviously, there is a predatory relationship between the Industrial System and the Technology Innovation System. However, industrial output has a greater impact on technological innovation, so technological progress must be taken seriously.

(3) Hebei area

Through Equation (34), one can find that the value in Industrial System is negative, and the value in Technology Innovation System is less than 1. This value of expresses the parameter of the system itself, and the value is generally greater than 1. However, the data in Hebei showed a negative value. This means that there are serious problems in the industrial system, which need to be improved.

In Table 5, β12=−0.00002646,β21=−0.00454712, indicating that there is a competitive relationship between Water Resources System and Economy System. In addition, the influence degree of Economy System on Water Resources System is greater than the influence degree of Water Resources System on Economy System. The Economy System determines the degree of change in the Water Resources System. It can also be said that the level of consumption determines the degree of water consumption, but the growth of the economic system should not be based on sacrificing water resources. The goal wants to accomplish is to decrease the rational use of water resources while the economy grows steadily, so as to achieve a state of win-win cooperation.

From Table 5, $\beta_{13}>0,\beta_{31}>0$, a mutualistic relationship is formed between Water Resources System and Industrial System. $\beta_{14}>0,\beta_{41}>0$, a mutualistic relationship is formed between Water Resources System and Technology Innovation System. $\beta_{34}>0,\beta_{43}>0$, there is a symbiotic relationship between Industrial System and Technology Innovation System. The relationship between Water Resources System, Industrial System and Technology Innovation System is symbiosis, which shows that the current general trend in Hebei is conducive to the development of Technology Innovation System.

In accordance with Table 5**,**
$\beta_{23}>0,\beta_{32}<0$, $\beta_{24}>0,\beta_{42}<0$ show that the relationship between Economic System, Industrial System and Technology Innovation System in Hebei is predicated. Due to $\beta_{23}=0.00000089,\beta_{32}=-0.00011921$, the influence of the Industrial System on the Economic System is greater than the influence of the Economic System on the Industrial System. Hebei Province is a province with a large proportion of the secondary industry in the Beijing-Tianjin-Hebei region. In recent years, it has been consistently optimizing and upgrading its industrial structure. The upgrading of industrial structure will have a great impact on the economy, but there is a predatory relationship between Economic System and Technology Innovation System. The continuous progress of science and technology requires strong support from the economy. However, in turn, the advancement of science and technology will enhance the manufacturing capacity of industrial enterprises or provide suggestions for the optimization and upgrading of the industrial structure in the province. Technology Innovation System and Economic System should be a state of symbiosis or benign competition, rather than a relationship of predation. Therefore, Hebei Province should pay attention to the relationship between systems, so that the various system levels within the province can form a situation of balanced competition.

#### 4.2.2. Analysis of Equilibrium and Stability

The stability of the system is mainly analyzed according to the calculated values of p,q,qr−ps, and $qrs-ps^{2}$. Table 6, Table 7 and Table 8, respectively, show the stability calculation of Beijing, Tianjin and Hebei.

From Table 6 that $x_{1},x_{2},x_{3},x_{4}$ in E15 is greater than 0, but $p,q,qr-ps,qrs-ps^{2}$ in E15 has a value not greater than 0. $x_{1},x_{2},x_{3},x_{4}$ are not all greater than 0 from E0 to E14, and the range of values fluctuates greatly. It is believed that the four systems of Beijing Water Resources System, Economic System, Industrial System, and Technology Innovation System have not attained a trend of consistent competition and coordinated growth since 2015.

By calculating the stability of Tianjin data, it is found that the value of *p*, *q* in the calculated data at point E15 in Table 7 is less than 0. It shows that in Tianjin, Water Resources System, Economic System, Industrial System, and Technology Innovation System have not attained stable and coordinated development. However, the values of $p,q,qr-ps,qrs-ps^{2}$ at point E1, E2 are all greater than zero. It shows that the development of Water Resources System and Economic System in the region is consistently stable. The coordination between systems has not been achieved. It is still necessary to enhance the coordination capabilities between systems to mutually foster high-quality development.

In Table 8, Hebei is similar to Beijing and Tianjin, and there are unstable phenomena. In this table, only the E2 data in which $p,q,qr-ps,qrs-ps^{2}$ is greater than 0. However, the data of $x_{1},x_{2},x_{3},x_{4}$ in E2 only has $x_{2}$ greater than 0, and all other data is 0. It shows that the Economic System of Hebei Province has reached a trend of steady growth. However, Hebei lacks the ability to collaborate between systems.

#### 4.2.3. Analysis of Forecast Data

(1) Beijing area

Through the above data results, it is found that the relationship between the predicted data and the calculated system will have a lot of changes with the current situation. This phenomenon further proves that the current problems in the Beijing area require the establishment of more effective methods to promote stable competition between systems.

From Table 9 and Table 10, find that the relationship between the Water Resources System and the Economic System in Beijing has changed from the previous predation to an incompatible relationship. This phenomenon shows that there is a big problem between the Water Resources System and the Economic System. Thus, a good coupling relationship must be maintained between water conservation and stable economic development. Water Resources System and Technology Innovation System has changed from the previous competition to the current predator relationship, indicating that the development of one system between the two in the future will cause obstacles to the development of the other system. Other systems have developed in the direction of healthy competition, or in symbiosis. However, they must continue to improve and become better at maintaining this good posture.

(2) Tianjin area

According to the relationship between the systems in Table 11 and Table 12, find that the relationship between the five systems has changed. The Water Resources System and Economic System have changed from the prior predator relationship to the current competitive relationship. Water Resources System and Technology Innovation System have changed from a symbiotic relationship to a predator relationship. Economic System and Industrial System have changed from the previous competitive relationship to the predator relationship. Economic System and Technology Innovation System have changed from the previous predation to the current competitive relationship. The relationship between Industrial System and Technology Innovation System has changed from the previous predation to the current symbiotic relationship.

The constant changes in the relationship between systems confirm the previous stable conclusion. This phenomenon emergence is mostly because each system has not found a more suitable development direction and situation. This needs to fundamentally find a way to improve, to find the direction of coordinated development of each system.

(3) Hebei area

From Table 13 and Table 14, comparing the previous data, find that the relationship between the five systems has changed. The Water Resources System and Economic System have changed from the previous competitive relationship to a predator relationship. Water Resources System and Industrial System have changed from mutualistic to predation. Water Resources System and Technology Innovation System have changed from symbiosis to competition. Economic System and Industrial System changed from predation to symbiosis. Economic System and Industrial System have changed from predation to a mutualistic relationship.

Half of the relationship between systems in Hebei Province has become symbiosis, which refers to the mutual existence of two systems. This state seems to be in a stable situation, but as resource policies continue to change, the relationship between systems will also change. If the symbiosis relationship has been maintained or tends to a mutualistic state, it will undoubtedly cause problems between systems and affect the development of Hebei Province.

### 4.3. Discussion

#### 4.3.1. Research Implications

As the coordinated and integrated development of the Beijing-Tianjin-Hebei region is proposed, high-quality development among various regions has also become an issue that needs to study now. High-quality development is inseparable from technological innovation, but the level of technological innovation cannot fully represent whether the region meets high-quality. High-quality development means that all resources between various regions can achieve a reasonable allocation, and can coordinate with each other and promote together.

As the capital economic circle, the Beijing-Tianjin-Hebei region is a key research area. Han et al. [8] used the coupling method to analyze the Economy-Resources-Environment of the region. Hu et al. [10] conducted research on industrial energy and related emissions in the Beijing-Tianjin-Hebei region. Zhang et al. [27] investigated the competition and cooperation between Shaanxi Province’s TI-RE-EE-IQ by constructing a four-dimensional Grey Lotka-Volterra model, and explored the balance and stability of the four-dimensional model. On this basis, the Water Resources System-Economic System-Industrial System-Technological Innovation System is constructed, and the Beijing-Tianjin-Hebei region is analyzed and studied by multi-dimensional Grey Lotka-Volterra model.

In the Beijing-Tianjin-Heber regions, most scholars used the grey correlation method or the relationship of calculating the coupling degree of the system to analyze the relationship between the systems. Secondly, few scholars in the region have paid attention to the connection between technological innovation and industry. Moreover, few scholars would think of using the Grey Lotka-Volterra calculation model to analyze the relationship between systems. Although many scholars in Jing-Jin-Ji have put forward their own suggestions and methods for improvement, there are few analyses on the systems of the Beijing-Tianjin-Hebei region. Most scholars study the three aspects of energy, economy, and industry in the Beijing-Tianjin-Hebei, or the relationship between the three, and rarely involve the opposite aspects of technological innovation. Thus, analyzing the relationship among the Beijing-Tianjin-Hebei Water Resources System-Economic System-Industrial System-Technological Innovation System is of great research value. It not only fills the vacancies in this area of the Beijing-Tianjin-Hebei region but can also provide references for other areas in this area.

#### 4.3.2. Research Limitations

This study has certain limitations. In the first place, the phenomenon of agglomeration of manufacturing industries in the Beijing-Tianjin-Hebei region [33]. This article does not consider the impact of the concentration of industrial activities on water resources, the economy, and industry. In future research, we will consider the impact of industrial concentration on water resources and the impact of industrial agglomeration on other systems. Secondly, although have extended the GLV model to four dimensions and selected 12 indicators of the four systems. However, there are still limitations and there is still room for improvement.

## 5. Conclusions

This paper takes Beijing-Tianjin-Hebei region as the research area, and first builds a composite Grey Lotka-Volterra model among the Water Resources System-Economic System-Industrial System-Technology Innovation System. Analyze the relationship between the four systems from 2015 to 2019 through the model and calculate the stability of the relationship between the systems. Secondly, the FGM(1,1) model is used to forecast the 12 evaluation indicators under the four evaluation systems. After obtaining the forecast data, the compound GLV model is used to analyze the relationship between the systems on the forecast data.

Through the analysis of the Beijing-Tianjin-Hebei region, the following conclusions are obtained.

1.Interaction between current systems. The Economic System in Beijing has a positive influence on the development of the Water Resources System, and Technology Innovation System has a positive effect on the development of the Economic System and the Industrial System. In the Tianjin area, Economic System and Industrial System are factors hindering the development of Water Resources System, and the development of Technology Innovation System is influenced by Economic System and Industrial System. There are visible differences between Hebei, Beijing and Tianjin. In Hebei, there is a competitive relationship between the Water Resources System and the Economic System, and there is a symbiotic relationship between the Water Resources System and the Industrial System and Technology Innovation System.2.Stability analysis. The relationship between the systems in Beijing-Tianjin-Hebei region is extremely unstable, and the relationship between systems has not formed a steady growth trend.3.Predict interaction between systems. In Beijing, there is a state of incompatibility between the Water Resources System and the Economic System, which is seriously inconsistent with the trend of coordinated development between systems. As the Technology Innovation System in Beijing has changed over time, both Water Resources System and Industrial System have become obstacles to it. In Tianjin, the development of the Economic System and Industrial System will inevitably have to sacrifice the Water Resources System, and the Water Resources System has promoted the development of the Industrial System and the Economic System. In Hebei, the Water Resources System hinders the development of the Economic System and Industrial System. Economic System, Industrial System and Technology Innovation System promote each other and make progress together.

Through the analysis of the systems in Beijing-Tianjin-Hebei, it is found that the relationship between the system and the system is poor, and most of the relationship is unstable. At the same time, it is found that the Beijing-Tianjin-Hebei system lacks cohesion and is still far from achieving coordinated development.

Based on the results, this paper puts forward the following suggestions.

(1)Strengthen innovation incentive policies. This is the main way to promote the development of high-quality industries and reduce waste of resources.(2)Formulate more reasonable and complete policies for personnel training and introduction.(3)Colleges and universities should strengthen the relationship with enterprises. On the premise of expanding the application scope of university scientific research achievements and promoting economic growth, accelerating the development of high-quality industry and rational utilization of resources.(4)Industrial enterprises should actively participate in scientific and technological innovation research.

Future research can be improved by adding dimensions and deepening research depth. From the perspective of adding dimensions, time and space can be added for analysis, or new system indicators can be added. From the perspective of research depth, evaluation indicators can be added on this basis to make the research results more detailed and comprehensive, and more in line with local characteristics.

## Figures and Tables

**Figure 1 ijerph-19-08969-f001:**
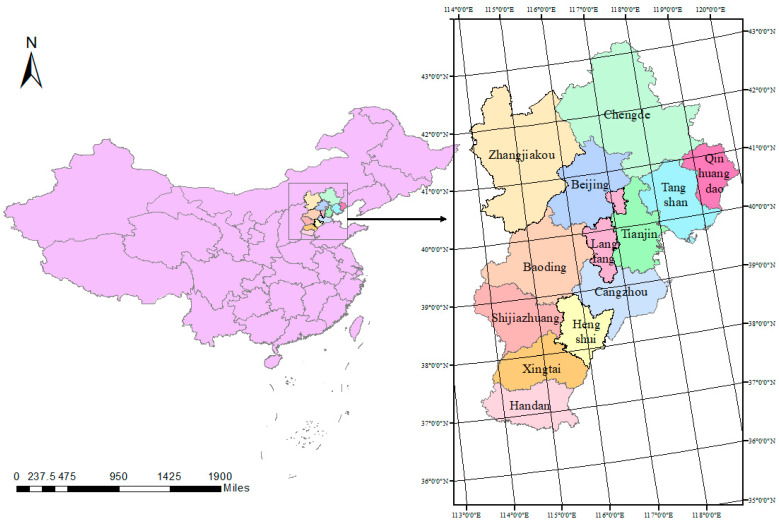
The geographical location of the Beijing-Tianjin-Hebei. (Refer to the map of the People’s Republic of China in http://www.gov.cn, accessed on 1 May 2022).

**Figure 2 ijerph-19-08969-f002:**
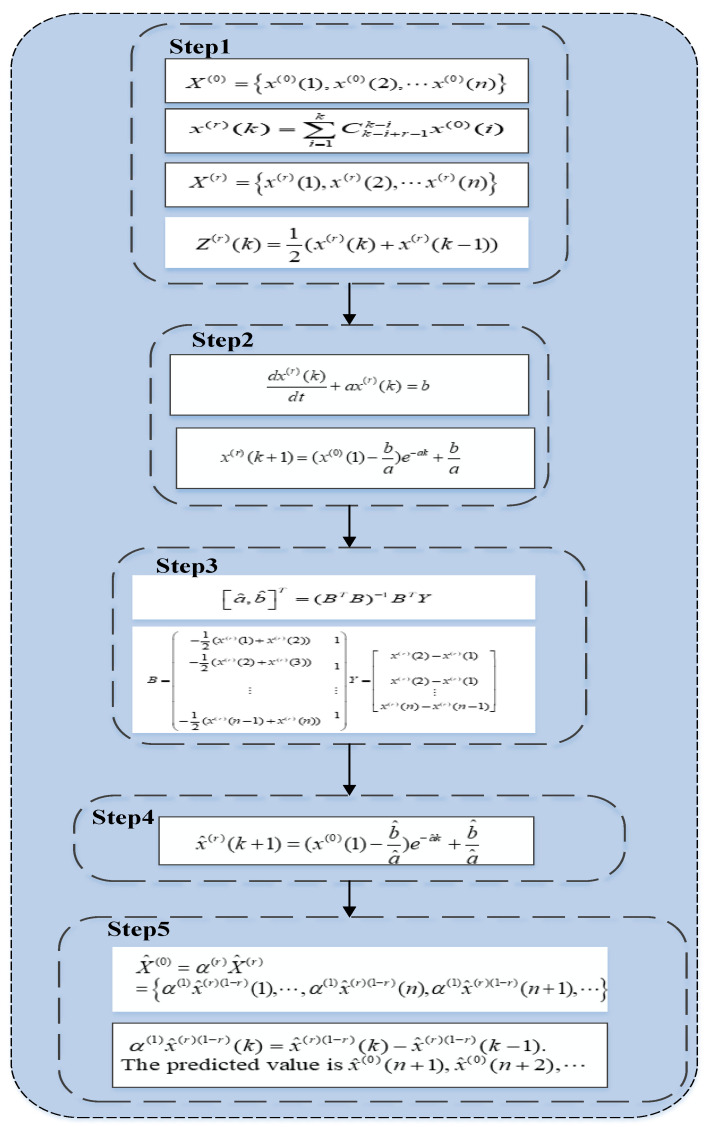
Framework diagram of FGM(1,1).

**Figure 3 ijerph-19-08969-f003:**
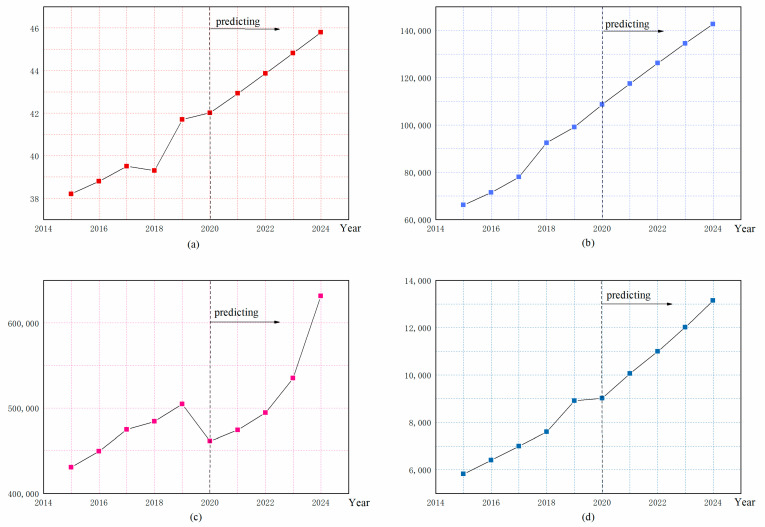
Forecasts of various systems in Beijing (**a**) WR system forecast; (**b**) ES system forecast; (**c**) IS system forecast; (**d**) TI system forecast.

**Figure 4 ijerph-19-08969-f004:**
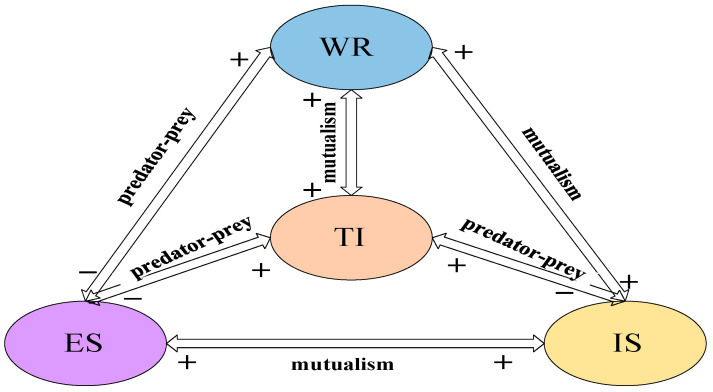
The relationship between Water Resources System-Economy System-Industrial System-Technology Innovation System in Beijing.

**Figure 5 ijerph-19-08969-f005:**
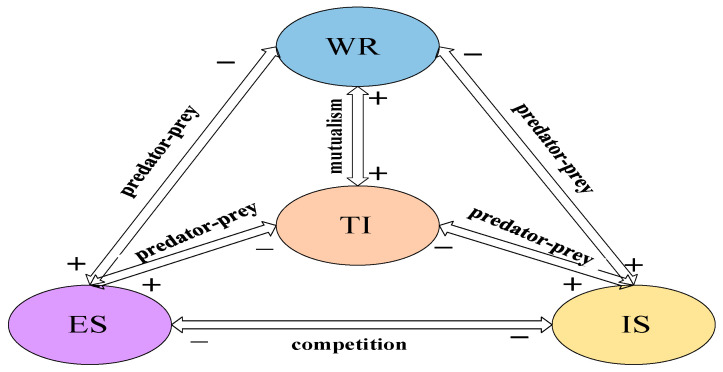
The relationship between Water Resources System-Economy System-Industrial System-Technology Innovation System in Tianjin.

**Figure 6 ijerph-19-08969-f006:**
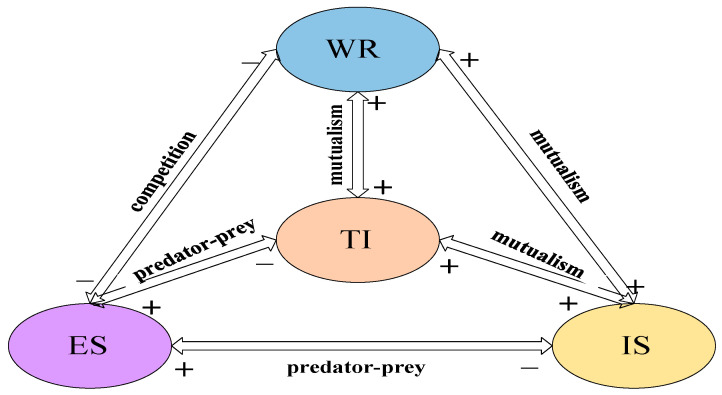
The relationship between Water Resources System-Economy System-Industrial System-Technology Innovation System in Hebei.

**Table 1 ijerph-19-08969-t001:** Evaluation index.

Evaluation System	First Level Indicator	Secondary Indicators
Water Resources System	Water system consumption	Total water supply (100 million cubic meters)
Economic System	Economic system output	Gross Regional Product (100 million yuan)
	Economic system output per capita	GDP per capita (yuan)
Industrial System	Output of each industry	Added value of primary industry (100 million yuan)
		Industrial added value (100 million yuan)
		Added value of construction industry (100 million yuan)
		Added value of tertiary industry (100 million yuan)
Technology Innovation System	Technological innovation investment	Internal expenditure of R&D funds (ten thousand yuan)
	Technological innovation investment	Full-time equivalent of R&D personnel (person-years)
	Scientific and technological innovation output	Number of valid invention patents granted (pieces)
		Number of patent applications (pieces)
		Invention patent (pieces)

**Table 2 ijerph-19-08969-t002:** Interaction between the various factors.

*β_ij_*	*β_ij_*	Interspecific Relationship
<0	<0	Competition
>0	>0	Mutualism
=0	=0	No interaction
>0	<0	Population *i* predator population *j*
<0	>0	Population *j* predator population *i*
<0	=0	Amensalism

**Table 3 ijerph-19-08969-t003:** Related parameters of GLV model in Beijing area.

${\hat{\beta }}_{ij}(i,j=1,2,3,4)$	WR(j=1)	ES(j=2)	IS(j=3)	TI(j=4)
WR(i=1)	−0.04638672	0.00001788	0.00000191	−0.00009537
ES(i=2)	−0.01611328	0.00003874	0.00000042	−0.00040436
IS(i=3)	0.04296875	0.00003815	−0.00000620	−0.00033569
TI(i=4)	−0.04394531	0.00001955	0.00000155	−0.00011063

**Table 4 ijerph-19-08969-t004:** Related parameters of GLV model in Tianjin area.

${\hat{\beta }}_{ij}(i,j=1,2,3,4)$	WR(j=1)	ES(j=2)	IS(j=3)	TI(j=4)
WR(i=1)	0.04351807	−0.00004959	−0.00005177	0.00082397
ES(i=2)	0.02166748	0.00000000	−0.00004460	0.00013733
IS(i=3)	0.03549194	−0.00001907	−0.00005405	0.00039673
TI(i=4)	0.02306366	−0.00001386	−0.00004627	0.00038528

**Table 5 ijerph-19-08969-t005:** Related parameters of GLV model in Hebei area.

${\hat{\beta }}_{ij}(i,j=1,2,3,4)$	WR(j=1)	ES(j=2)	IS(j=3)	TI(j=4)
WR(i=1)	−0.00848389	−0.00002646	0.00000030	0.00023961
ES(i=2)	−0.00454712	−0.00001383	0.00000089	0.00003588
IS(i=3)	0.01733398	−0.00011921	0.00000108	0.00003242
TI(i=4)	0.00531006	−0.00009012	0.00000040	0.00024438

**Table 6 ijerph-19-08969-t006:** Estimated equilibrium and eigenvalues in Beijing.

*E*	Equilibrium				Stability			
	$x_{1}$	$x_{2}$	$x_{3}$	$x_{4}$	p	q	qr−ps	$qrs-ps^{2}$
E0	0.0	0.0	0.0	0.0	−3.2	3.5	−1.3	0.1
E1	17.0	0.0	0.0	0.0	−1.3	−0.2	0.8	−0.2
E2	0.0	−30,902.7	0.0	0.0	1.5	−0.4	−0.8	0.2
E3	0.0	0.0	10,082.5	0.0	−3.1	3.2	−0.9	−0.1
E4	0.0	0.0	0.0	10,522.5	7.9	19.9	16.2	2.6
E5	6.1	−28,367.2	0.0	0.0	1.8	0.6	−0.2	−0.1
E6	24.4	0.0	179,222.0	0.0	1.0	−1.6	−0.4	0.3
E7	−25.1	0.0	0.0	20,495.2	15.7	68.4	51.2	−25.6
E8	0.0	−29,083.9	−168,895.2	0.0	−0.2	−1.3	0.4	0.0
E9	0.0	−93,456.1	0.0	−5993.5	4.8	7.8	5.0	0.9
E10	0.0	0.0	−318,289.5	6063.7	0.5	−4.1	−4.9	−1.3
E11	−2.2	−29,788.7	−188,497.9	0.0	−0.5	−1.4	0.6	0.0
E12	−78.1	−438,385.6	0.0	−35,930.7	17.3	82.7	45.9	−140.4
E13	535.2	0.0	8,337,691.8	−85,273.8	36.5	−2031.1	−180.3	−10,350.0
E14	0.0	−61,600.8	−198,618.6	−3146.2	1.2	−1.4	−1.7	−0.4
E15	749.6	1,923,546.5	8,069,200.8	165,717.4	28.6	−2023.9	1827.5	−27,265.9

**Table 7 ijerph-19-08969-t007:** Estimated equilibrium and eigenvalues in Tianjin.

E	Equilibrium				Stability			
	$x_{1}$	$x_{2}$	$x_{3}$	$x_{4}$	p	q	qr−ps	$qrs-ps^{2}$
E0	0.00	0.00	0.00	0.00	−3.60	3.93	−1.23	0.00057
E1	−41.47	0.00	0.00	0.00	2.43	1.23	0.16	0.00003
E2	0.00	166,229.33	0.00	0.00	10.11	27.04	21.71	0.01008
E3	0.00	0.00	23,236.64	0.00	1.19	−0.41	−0.40	−0.00001
E4	0.00	0.00	0.00	−1410.06	−0.80	−0.28	0.24	−0.00009
E5	−26.80	12,875.43	0.00	0.00	1.36	0.21	−0.02	0.00000
E6	−63.19	0.00	−18,261.65	0.00	1.83	−0.47	−0.04	−0.00001
E7	110.72	0.00	0.00	−8037.63	−3.72	5.43	−3.98	0.00662
E8	0.00	4232.19	21,743.06	0.00	1.23	−0.18	−0.29	0.00001
E9	0.00	1,469,690.95	0.00	51,452.43	15.23	−512.10	−3617.95	−1.31941
E10	0.00	0.00	108,868.52	11,665.67	−4.39	−11.17	18.13	0.01766
E11	−18.42	13,613.70	6337.69	0.00	1.51	0.48	0.02	0.00000
E12	−30.16	40,992.43	0.00	1869.93	0.45	0.04	−0.02	−0.00003
E13	−71.52	0.00	−22,418.32	178.59	1.83	−0.79	0.06	0.00001
E14	0.00	108,460.20	27,423.08	5784.75	−0.52	−0.56	−0.09	0.00009
E15	16.20	128,000.59	42,488.54	7327.27	−1.23	−1.39	1.00	−0.00020

**Table 8 ijerph-19-08969-t008:** Estimated equilibrium and eigenvalues in Hebei.

E	Equilibrium				Stability			
	$x_{1}$	$x_{2}$	$x_{3}$	$x_{4}$	p	q	qr−ps	$qrs-ps^{2}$
E0	0	0	0	0	−3	2	1	−1
E1	179	0	0	0	−3	0	8	−4
E2	0	105,225	0	0	25	177	355	213
E3	0	0	658,072	0	−5	11	−9	3
E4	0	0	0	−3097	−1	−2	0	1
E5	5790	−1,798,559	0	0	−483	48,520	1,458,213	1,911,593
E6	130	0	−1,423,023	0	2	1	−2	−1
E7	57	0	0	−4332	−1	−1	−1	0
E8	0	−24,278	−2,020,869	0	−2	1	−10	15
E9	0	2,256,099	0	828,908	−69	−51,906	5,814,759	9,281,706
E10	0	0	790,158	−4401	−2	0	3	−1
E11	132	5257	−881,967	0	2	0	−1	0
E12	248	42,182	0	7079	2	−5	−13	−7
E13	58	0	−148,262	−4111	1	0	−1	0
E14	0	−23,285	−1,641,980	−8975	4	9	11	4
E15	−591	−152,286	−5,542,691	−37,266	8	7	48	−337

**Table 9 ijerph-19-08969-t009:** GLV model related parameters of forecast data in Beijing area.

${\hat{\beta }}_{ij}(i,j=1,2,3,4)$	WR(j=1)	ES(j=2)	IS(j=3)	TI(j=4)
WR(i=1)	−0.04296875	0.00000000	−0.00000346	0.00039673
ES(i=2)	−0.04589844	0.00000358	−0.00000325	0.00026321
IS(i=3)	−0.03466797	−0.00000954	−0.00000343	0.00038910
TI(i=4)	−0.06591797	0.00002289	−0.00000274	0.00010872

**Table 10 ijerph-19-08969-t010:** The relationship between forecasting data systems in Beijing area.

Index	Numerical Value	Index	Numerical Value	Interspecific Relationship
β12	0.00000000	β21	−0.04589844	Amensalism
β13	−0.00000346	β31	−0.03466797	Competition
β14	0.00039673	β41	−0.06591797	Predation
β23	−0.00000325	β32	−0.00000954	Competition
β24	0.00026321	β42	0.00002289	Mutualism
β34	0.00038910	β43	−0.00000274	Predation

**Table 11 ijerph-19-08969-t011:** GLV model related parameters of forecast data in Tianjin area.

${\hat{\beta }}_{ij}(i,j=1,2,3,4)$	WR(j=1)	ES(j=2)	IS(j=3)	TI(j=4)
WR(i=1)	−0.97167969	−0.00100517	0.00226927	0.00622559
ES(i=2)	−0.71972656	−0.00021553	0.00195122	−0.00492859
IS(i=3)	−0.88671875	−0.00072479	0.00216007	0.00230408
TI(i=4)	−1.05126953	−0.00115395	0.00241613	0.00790405

**Table 12 ijerph-19-08969-t012:** The relationship between forecasting data systems in Tianjin area.

Index	Numerical Value	Index	Numerical Value	Interspecific Relationship
β12	−0.00100517	β21	−0.71972656	Competition
β13	0.00226927	β31	−0.88671875	Predation
β14	0.00622559	β41	−1.05126953	Predation
β23	0.00195122	β32	−0.00072479	Predation
β24	−0.00492859	β42	−0.00115395	Competition
β34	0.00230408	β43	0.00241613	Mutualism

**Table 13 ijerph-19-08969-t013:** GLV model related parameters of forecast data in Hebei area.

${\hat{\beta }}_{ij}(i,j=1,2,3,4)$	WR(j=1)	ES(j=2)	IS(j=3)	TI(j=4)
WR(i=1)	−0.00488281	0.01002502	0.00001642	−0.04504395
ES(i=2)	−0.51635742	0.00070381	0.00000510	0.00648499
IS(i=3)	−0.44720459	0.00182343	0.00000642	0.00013733
TI(i=4)	−0.52215195	0.00057882	0.00000494	0.00714779

**Table 14 ijerph-19-08969-t014:** The relationship between forecasting data systems in Hebei area.

Index	Numerical Value	Index	Numerical Value	Interspecific Relationship
β12	0.0100250244	β21	−0.5163574219	Predation
β13	0.0000164211	β31	−0.4472045898	Predation
β14	−0.0450439453	β41	−0.5221519470	Competition
β23	0.0000051018	β32	0.0018234253	Mutualism
β24	0.0064849854	β42	0.0005788207	Mutualism
β34	0.0001373291	β43	0.0000049373	Mutualism

## Data Availability

The research data in this article comes from Beijing Statistical Yearbook, Tianjin Statistical Yearbook, and Hebei Economic Yearbook over the years.

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
