# Peer review of "Application of Grey Lotka-Volterra Model in Water-Economy-Industry-Technology Innovation System in Beijing-Tianjin-Hebei Region"

_ijerph, 2022, doi:10.3390/ijerph19158969_

Round 1

Reviewer 1 Report

Dear authors,

I congratulate you on your chosen topic.

My main suggestions are as follows:

I recommend inserting research hypothesis(es) to be validated or invalidated later.

I recommend inserting a discussion section before the conclusions in order to discuss the results of the empirical research in a more simplistic manner, so that the main results can be much clearer.

Author Response

I recommend inserting research hypothesis(es) to be validated or invalidated later.

Reasonable modifications have been made. In the paper model introduction and calculation results of the corresponding part of the hypothesis.

I recommend inserting a discussion section before the conclusions in order to discuss the results of the empirical research in a more simplistic manner, so that the main results can be much clearer.

The structure of the paper has been readjusted and 4.3 discussion has been added before 5. Conclusion

Reviewer 2 Report

Dear authors,

The manuscript entitled “Application of Grey Lotka-Volterra Model in Water-Economy- Industry-Technology Innovation System in Beijing-Tianjin- Hebei Region” by Xiaorui Guo et al covers an interesting topic focused on the composite Grey Lotka-Volterra (GLV) model and Fractional GM(1,1)(FGM(1,1)) model in order to analyze the relationship between data of Water Resources System-Economic System-Industrial System-Technology Innovation System in Beijing-Tianjin-Hebei region from 2015 to 2019. The paper includes some interesting results which may be useful in providing direction for the future development of Beijing-Tianjin- Hebei region and serve as references for other regions too. My personal opinion is that the paper could be of interest to other researchers but I have some concerns regarding the following points:

11.Firstly, the manuscript doesn’t follow the structure of a regular research article as described in Instructions for authors. I recommend that the authors restructure their paper according to  journal’s instructions for a research manuscript (Introduction, Materials and Methods, Results, Discussion, Conclusion). I think that by doing so the manuscript could be easier to follow and understand. For example the author could comprise Section 1 (Introduction) and 2 (Literature overview) just in one Section – Introduction. The results section should make a clear presentation of the results achieved by the research and all the comments and discussions regarding the results should be introduce in a dedicated section for Discussion (which in this form of the manuscript is missing). The Conclusion section should be the last section of the manuscript and more synthetic from my point of view. Section 7 could be introduced in discussion section.

22. Secondly, I recommend that the authors review the Introduction and introduce in this section some information’s regarding the actual state of water resources in China/ and or the regions studied.

33.       Thirdly, I recommend that the authors perform a review in terms of English language and spelling as well as regarding the scientific soundness (which could be improve because some parts of the manuscript look more like a guideline rather than a research article). I have listed below some examples of such phrases which I recommend to be reformulated:

 -              Some references are not properly cited because the authors specified only the name of the author cited without specifying the number of the reference. See Lines : 58 (“Yang et al.”);  60:“Han et al. ”; Line 62 – “Tang et al.”; Line 65 – “Hu et al.”; Line 68 - Xu et al. etc

- Lines 76-80 – I recommend to review the following phrases - “Therefore, in order to make up for the lack of research on the relationship between the Beijing-Tianjin-Hebei region systems. This paper decides to use the Grey Lotka-Volterra model to study the Water Resources System-Economy System-Industrial System-Technology Innovation System in Beijing-Tianjin-Hebei and explore the competition and cooperation between the four systems.”

- Lines 90-91 – I recommend to review the following phrase- “Analyzed the synergistic effect of Wuhan's industrial upgrading and urban expansion and predicted the development status in the next five years [15].”

- Line 94: Please provide a normal font for “Lotka-Volterra” because in this current form of the manuscript this is superscrip

- Line 172: “Figure 2 below,” – the comma should be replaced with a point “.”

- The equations from sub-section 4.3.1 should be formatted according to the instructions for authors (and numbered)

- Line 251 – please check the font for the following words “p, q, r” as they seem different from the original font

- Line 263: “We assumption that” I recommend to be replaced with “We assume that/ Our assumption is”

-  268: “”Where, WR is Water Resources System, ES is Economic System, IS is Industrial System and TI is Technology Innovation System.”- This phrase looks incomplete

- Line 273: “Combine the formulas in the fourth part to calculate and organize the following formulas.”

- Line 287: “It can be seen from Table 3 that reflects the existence”

- The expressions “Because of,” and “it can be seen” are used very often and sometimes inappropriate from my point of view thus I recommend that authors try to replace these with another synonym more suitable and with scientific soundness. See Lines 314, 324 (for “Because of,” ) and Lines 287, 298, 317, 358, 367, 372, 381 etc (for“it can be seen”).

Author Response

Detailed Response to Reviewer

Firstly, thanks for your careful and insightful comments. The revisions are listed as follows:

Firstly, the manuscript doesn’t follow the structure of a regular research article as described in Instructions for authors. I recommend that the authors restructure their paper according to journal’s instructions for a research manuscript (Introduction, Materials and Methods, Results, Discussion, Conclusion). I think that by doing so the manuscript could be easier to follow and understand. For example, the author could comprise Section 1 (Introduction) and 2 (Literature overview) just in one Section – Introduction. The results section should make a clear presentation of the results achieved by the research and all the comments and discussions regarding the results should be introduce in a dedicated section for Discussion (which in this form of the manuscript is missing). The Conclusion section should be the last section of the manuscript and more synthetic from my point of view. Section 7 could be introduced in discussion section.

The overall structure of the article has been reasonably adjusted, according to journal’s instructions for a research manuscript (Introduction, Materials and Methods, Results, Discussion, Conclusion). Section 1 and Section 2 are integrated, and the position of Section 7 is adjusted.

Secondly, I recommend that the authors review the Introduction and introduce in this section some information’s regarding the actual state of water resources in China/ and or the regions studied.

The introduction of the article has been adjusted, adding the introduction of research indicators such as water resources and the studied region.

Thirdly, I recommend that the authors perform a review in terms of English language and spelling as well as regarding the scientific soundness (which could be improve because some parts of the manuscript look more like a guideline rather than a research article). I have listed below some examples of such phrases which I recommend to be reformulated:

 - Some references are not properly cited because the authors specified only the name of the author cited without specifying the number of the reference. See Lines : 58 (“Yang et al.”);  60:“Han et al. ”; Line 62 – “Tang et al.”; Line 65 – “Hu et al.”; Line 68 - Xu et al. etc

- Lines 76-80 – I recommend to review the following phrases - “Therefore, in order to make up for the lack of research on the relationship between the Beijing-Tianjin-Hebei region systems. This paper decides to use the Grey Lotka-Volterra model to study the Water Resources System-Economy System-Industrial System-Technology Innovation System in Beijing-Tianjin-Hebei and explore the competition and cooperation between the four systems.”

- Lines 90-91 – I recommend to review the following phrase- “Analyzed the synergistic effect of Wuhan's industrial upgrading and urban expansion and predicted the development status in the next five years [15].”

- Line 94: Please provide a normal font for “Lotka-Volterra” because in this current form of the manuscript this is superscrip

- Line 172: “Figure 2 below,” – the comma should be replaced with a point “.”

- The equations from sub-section 4.3.1 should be formatted according to the instructions for authors (and numbered)

- Line 251 – please check the font for the following words “p, q, r” as they seem different from the original font

- Line 263: “We assumption that” I recommend to be replaced with “We assume that/ Our assumption is”

-  268: Where, WR is Water Resources System, ES is Economic System, IS is Industrial System and TI is Technology Innovation System.”- This phrase looks incomplete

- Line 273: “Combine the formulas in the fourth part to calculate and organize the following formulas.”

- Line 287: “It can be seen from Table 3 that reflects the existence”

- The expressions “Because of,” and “it can be seen” are used very often and sometimes inappropriate from my point of view thus I recommend that authors try to replace these with another synonym more suitable and with scientific soundness. See Lines 314, 324 (for “Because of,”) and Lines 287, 298, 317, 358, 367, 372, 381 etc (for“it can be seen”).

It has been modified. Spelling check and grammar revision are carried out on the whole paper. Especially the examples given.

We would appreciate it if you have any question or advice about this paper. Please don't hesitate to let us know.

Thank you again for your constructive and encouraging comments.

Reviewer 3 Report

This paper applied composite Grey Lotka-Volterra (GLV) model and Fractional GM(1,1)(FGM(1,1)) model to analyze the relationship between data of Water Resources System-Economic System-Industrial System-Technology Innovation System in Beijing-Tianjin-Hebei region from 2015 to 2019. Overall, this study addresses a topic of high relevance for research and also for practice. However, I believe some issues need revision and clarification. 

Author Response

Detailed Response to Reviewer

Firstly, thanks for your careful and insightful comments. The revisions are listed as follows:

General comments

  1. The English grammar and style should be checked throughout the manuscript and specially in the Abstract.

The grammar and language style of the full text are revised.  In addition, the abstract part is mainly modified.

  1. The authors should avoid using pronouns such as “we”, “our” and “us” in the text.

The full text has been reviewed and revised to try to avoid such expressions.

  1. Given that the study presents a long list of abbreviations, I suggest adding a “glossary” table at the end of the paper as it will aid the readers to learn about the concepts/terms that they are about to study.

It has been modified. A long list of abbreviations listed in the article are sorted out and introduced at the end of this paper.

Abstract

  1. The authors should mention the main aim of the study in the Abstract section.

The abstract has been modified and the introduction of research objectives has been added.

  1. Lines 18-21 in page 1: “At the same time, through the stability calculation, it is found that the calculation results of most data are below 0, indicating that the Beijing-Tianjin-Hebei region Water Resources System Economic System-Industrial System-Technology Innovation System has not reached the relationship of mutual promotion and coordinated, and is in a state of unstable development.”is long and heavy; the authors should split it.

It has been modified. Has been modified into a simple short sentence, easy to read.

  1. I miss more emphasis on the main significance of this study in Abstract. I suggest

highlighting the main significance of the study in 1-2 sentences.

The abstract has been modified and the introduction of research significance has been added.

  1. The authors should mention a few policy implications after the main recommendation

based of results at the end of the Abstract in 1-2 sentences.

The abstract has been modified and the introduction of policy implication has been added.

  1. The authors should avoid repeating keywords already exists in the title (e.g. Grey Lotka-Volterra; Beijing-Tianjin-Hebei). The authors should replace them with new relevant words in the text.

It has been modified.

Introduction and Literature Review

  1. The Introduction section should be enriched by adding and citing several recent references (i.e. 2017- 2021). Also, the old references (1926-2005) should be replaced with the recent ones in the Introduction section as well as the whole manuscript.

It has been modified. References have been replaced and added.

  1. The Introduction section is too short and should be enriched by discussion on the main topic of the study.

It has been modified. The Introduction section has been expanded and modified.

  1. Inputs in the Introduction section should be enriched with recent and relevant references.

It has been modified. Recent and relevant references have been added in the Introduction section.

  1. In the Introduction section, there should be a paragraph discussing the global novelty of the study comparing with previous studies. This is very important to first identify the gap in the previous studies, and then highlight how the current study is going to fill it.

It has been modified. It has been added and revised in the last paragraph of Introduction section.

  1. Lines 58-60 in page 2: “In terms of rural and urban development, Yang et al. used a method of coupling and coordination to spatially analyze the relationship between production, life and ecology in the rural areas of Beijing-Tianjin- Hebei [6].The authors should modify the citation in the sentence to Yang et al. [6]. There are lots of such mistake; please check the whole paper and revise this issue.

It has been modified.

Methodology

  1. In the“3.1 Study Areas”subsection, the authors should add description about geographical and demographic information of the case study.

It has been modified. A description of the selected region has been added in this section.

  1. The authors should add source for the figure 1.

Refer to the map of the People's Republic of China in http://www.gov.cn

  1. The authors should add source for the Table 1.

The evaluation indexes of this paper were screened by referring to the data indexes in the provincial statistical yearbooks on the official website of the National Bureau of Statistics.

  1. Line 148 in page 5: The “4. Model Introduction” section should be a subsection of

Methodology. Therefore, the authors should modify it to 3.2. Model Introduction

It has been modified. The order of the full paper has been adjusted.

  1. The authors should cite all equations in the main text and define them.

It has been modified.

Results

  1. The authors mixed methodology and results. I suggest the authors add a section as “4. Results” and add all findings under this section. Moreover, all equations should be

to the Methodology section.

It has been modified. Reasonable separation of methodology and results.

Discussion

  1. Why the authors’ didn’t add the discussion section? In this section, authors should summarize the results and outline their interpretation in light of the published literature. Then, please explain the importance of the results and finally acknowledge the

shortcomings of the study.

It has been modified. The article has been revised and the discussion has been added in section 4.3

Conclusion

  1. The authors should highlight the specific and practical suggestions with respect to their findings at the end of the Conclusion section in one paragraph.

It has been modified.

  1. The authors should address future research direction at the end of the Conclusions section as well.

It has been modified. The order of conclusion was adjusted and the description of future research direction was added.

We would appreciate it if you have any question or advice about this paper. Please don't hesitate to let us know.

Thank you again for your constructive and encouraging comments.

Round 2

Reviewer 2 Report

Dear authors,

I read the new version of your manuscript and despite that I recognize that significant improvement have been provided, there are still some aspects which I recommend you should consider prior your manuscript publication.

I recommend that authors pay more attention to the English language and spelling because there are a lot of phrases which are not properly formed or which don't make any sense. I listed bellow some examples (but not all):

LINE 33 - " my country's society"

LINE 33-34 - "To achieve the coordination and unity of various institutional elements and promote higher-level development"

line 49-50 - "In March 2021, accelerate the coordinated development of the Beijing-Tianjin-Hebei region"

line 52-53 - "Therefore, in this context, the coordination of this region is calculated. "

line 165-167 - "Assumed that GLV model can accurately calculate and analyze the relationship between system, and the calculated data has a small error, which can accurately judge the relationship between system. "

line 279-280 - "Suppose there is a relationship between the selected research systems. Selected data are true and valid, and the calculation results and process are accurate."

line 338-339 - " So as to the good competition among all systems in Beijing-Tianjin-Hebei region and further foster the coordinated development between regions."

line 354-357 - "1), 2), 3)" - I think this is a typing error

line 361-362 - "From Figure 4 that there is a predator-predation relationship between the Economy System, Industrial System and Technology Innovation System. "

line 383 - "According to Table 3, indicates that there is a competitive relationship"

line 415 - "In Figure 5, know the internal connection between the four systems in Tianjin"

line 461 - "Combining Figure 6 and Table 5 to analyze the relationship between the four systems in Hebei."

I suggest that authors try to avoid using the first person statement and provide a more scientifically sound formulation (for example the use of "we/I" should be replaced); line 365 - "As you can see"

I recommend that authors provide a better resolution for figures 1 and 3.

Author Response

Firstly, thanks for your careful and insightful comments. The revisions are listed as follows:

Reviewer 2

I recommend that authors pay more attention to the English language and spelling because there are a lot of phrases which are not properly formed or which don't make any sense. I listed bellow some examples (but not all):

LINE 33 - " my country's society"

LINE 33-34 - "To achieve the coordination and unity of various institutional elements and promote higher-level development"

line 49-50 - "In March 2021, accelerate the coordinated development of the Beijing-Tianjin-Hebei region"

line 52-53 - "Therefore, in this context, the coordination of this region is calculated. "

line 165-167 - "Assumed that GLV model can accurately calculate and analyze the relationship between system, and the calculated data has a small error, which can accurately judge the relationship between system. "

line 279-280 - "Suppose there is a relationship between the selected research systems. Selected data are true and valid, and the calculation results and process are accurate."

line 338-339 - " So as to the good competition among all systems in Beijing-Tianjin-Hebei region and further foster the coordinated development between regions."

line 354-357 - "1), 2), 3)" - I think this is a typing error

line 361-362 - "From Figure 4 that there is a predator-predation relationship between the Economy System, Industrial System and Technology Innovation System. "

line 383 - "According to Table 3, indicates that there is a competitive relationship"

line 415 - "In Figure 5, know the internal connection between the four systems in Tianjin"

line 461 - "Combining Figure 6 and Table 5 to analyze the relationship between the four systems in Hebei."

I suggest that authors try to avoid using the first person statement and provide a more scientifically sound formulation (for example the use of "we/I" should be replaced); line 365 - "As you can see"

Sentences has been modified or deleted. Spelling check and grammar revision are carried out on the whole paper. Especially the examples given.

I recommend that authors provide a better resolution for figures 1 and 3.

It has been modified. Descriptions have been added to both Figures 1 and Figures 3.

Reviewer 3 Report

The authors have addressed my comments carefully and my major remarks from the previous version are addressed sufficiently. However, there are some minor points that need to be addressed. 

Author Response

Detailed Response to Reviewer

Firstly, thanks for your careful and insightful comments. The revisions are listed as follows:

Reviewer 3

  1. The English grammar and style should be checked significantly throughout the manuscript.

The grammar and language style of the full text have been checked and revised.

  1. The authors should mention the main aim of the study in the Abstract section.

The abstract has been modified and the introduction of research objectives has been added.

  1. The authors should add some policy implication to the end of the Abstract section.

The abstract has been modified and the introduction of policy implication has been added.

  1. The resolution of the Figure 1 (The geographical location of the Beijing-Tianjin-Hebei) is too low and the authors should modify it.

It has been modified. Figure 1 is redrawn.

  1. The authors should add definition for all equations.

It has been modified. The formula of the full text is numbered.

  1. Inputs of the Figure 2 is not readable; please modify it.

It has been modified.

  1. The explanations of Tables 5 to 8 is not enough and the authors should explain their findings with more details.

It has been modified. In this paper, the explanation of Table 5-8 is added.

  1. In line 277 in page 12: The authors should change “4. Case Analysis” to “4. Results and Discussion”.

It has been modified.

  1. Discussions on the main findings need further enrichment by comparing them with the existing literature. There should be some indications on how the main findings are in line with previous studies.

It has been modified. The Discussion section has been revised to add this description.

We would appreciate it if you have any question or advice about this paper. Please don't hesitate to let us know.

Thank you again for your constructive and encouraging comments.
